# Cognitive simplicity drives collective route improvements in homing pigeons

**Shoubhik Chandan Banerjee\*, Fritz A Francisco, Albert B Kao\***

Department of Biology, University of Massachusetts Boston, Boston, United States

## eLife Assessment

This study addresses an **important** question and shows how social navigation in homing pigeons can be explained by simple averaging, without requiring any complex cognitive abilities. The evidence, based on a rigorous and systematic comparison of seven models and data on how social routes can be generated from solitary routes, is **compelling**. The authors should be commended for their willingness to critically re-examine established interpretations.

**\*For correspondence:**
shoubhik.banerjee001@umb.edu
(SCB);
albert.kao@umb.edu (ABK)

**Competing interest:** The authors declare that no competing interests exist.

## Abstract

Cognitive abilities are central to how animals navigate complex environments. Beyond individual cognition, group living can also enhance navigation by pooling individually acquired information. One way this may be achieved is by following experienced leaders, which requires recognizing expertise within group members. Alternatively, accurate decisions could also emerge without expert opinions, through simpler mechanisms like the 'wisdom of crowds' principle that average out individual biases. Consequently, collective navigation strategies range from cognitively complex to simple, and yet, the prevalence or interplay of different collective strategies in nature remains unexplored. In this study, we asked: what is the navigation mechanism, requiring minimal cognitive demands, that is necessary and sufficient to quantitatively replicate the experimental results of a 2017 study on homing pigeons (*Columba livia*), which showed that sequential chains of bird pairs flying home—similar to a game of telephone—led to shorter homing routes compared to control birds flying individually or in fixed pairs. Our results show that the experimental data aligns closely with the simplest strategy—route averaging. Surprisingly, the complex mechanism of selectively propagating the best flight through social learning offered no additional advantage. We further observed that mixed strategies, although not supported by the experimental data, in theory combined advantages from both averaging and active selection of better routes, resulting in even greater performance. Hence, our results highlight the potential for future research to investigate selective pressures shaping the evolution of cultural learning and trade-offs among different decision mechanisms theoretically available to social animals in nature.

## Introduction

For decades, humans have been intrigued by the remarkable ability of animals to navigate their environment (*Krause and Ruxton, 2002*; *Wiltschko and Wiltschko, 2003*; *Åkesson et al., 2014*; *Wiltschko and Wiltschko, 2023*). Fascinating examples include the annual pole-to-pole migration of Arctic terns (*Egevang et al., 2010*; *Alerstam et al., 2019*) and sooty shearwaters (*Shaffer et al., 2006*) that achieve flight distances spanning thousands of kilometers or the precise navigation of anadromous salmon, swimming from the open ocean to return to their natal spawning grounds (*Thorpe, 1988*; *Quinn, 2018*). To achieve these marvelous feats, animals utilize a variety of environmental information to guide their journey at different length scales–for example, polarized light (*Muheim et al., 2006*) or geomagnetic cues (*Brothers and Lohmann, 2015*; *Karwinkel et al., 2024*) can roughly guide animals

across long length scales, while olfactory cues (*Bett and Hinch, 2016*) and visual landmarks (*Dreyer et al., 2018*) are more useful at shorter length scales.

Animals living in groups can further reduce directional uncertainty through social interactions. One way this can be achieved is by the direct social transfer of information. In such cases, the group leader (or a few members of the group, who are often older or more experienced) guide the rest of the group in their preferred direction via a leader-follower relationship (*Couzin et al., 2005*; *Nagy et al., 2010*; *Nagy et al., 2013*; *Strandburg-Peshkin et al., 2015*; *Strandburg-Peshkin et al., 2018*; *Sampaio et al., 2024*; *Berdahl et al., 2018*). For example, dominant female meerkats have higher influence in leading group movements, especially movement turn and speed, due to their experience and front-most position in the group (*Averly et al., 2022*). Through these leadership dynamics, expertise can propagate through the group, allowing less knowledgeable individuals to exhibit improved navigational performance.

An alternative socially-mediated mechanism that can improve migration efficiency is the 'wisdom of crowds', where large groups improve their decision by averaging individual opinions (*Berdahl et al., 2018*). This mechanism benefits from the law of large numbers, whereby individual errors cancel each other out with increasing group size, resulting in greater collective accuracy (*Galton, 1907*; *Simons, 2004*) (assuming that the distribution of opinions is roughly centered around the optimal value; *Kao et al., 2018*). In contrast to the leadership mechanism described above, here the group outcome is emergent. In other words, the collective decision may be highly accurate even if there are no knowledgeable individuals in the group at all. Furthermore, this mechanism has low cognitive requirements, since it does not require group members to recognize differences in knowledge or experience. In both humans and non-human animals, across different behavioral contexts, the wisdom of crowds has been shown to improve outcomes as group sizes increase (*Sumpter et al., 2008*; *Dell'Ariccia et al., 2008*; *Faria et al., 2009*; *Krause et al., 2011*; *Ward et al., 2011*; *Prelec et al., 2017*).

However, the two mechanisms described above are not mutually exclusive. Weighted average strategies (*Bahrami et al., 2010*; *Koriat, 2012*), where more experienced, knowledgeable, or confident individuals hold greater influence in the collective decision, but other individuals still have some decision weight, may capture the benefits of both existing knowledge and emergent solutions. While sociality can theoretically support a multitude of distinct and non-mutually exclusive strategies for improving navigation efficiency, the extent to which animals use each strategy in practice remains unclear. In particular, how the optimal strategy depends on the type of decision that animals face, the cognitive ability of the animals, and other details of the species' life history, remains poorly understood.

## Learning through time

Many animals usually make repeated trips between two locations in a given period of time. At small spatial scales, these journeys may be between known locations within their home range (*Strandburg-Peshkin et al., 2015*), or at larger spatial scales, they may be seasonal migrations that occur annually (*Flack et al., 2018*). A solitary individual may improve its performance across bouts by sampling multiple routes and preferentially remembering better ones. However, the individual faces a classic exploration-exploitation trade-off (*Mehlhorn et al., 2015*) between spending time exploring new routes (which may be associated with unknown and possibly severe risks) or utilizing previously known routes that are less costly. Furthermore, an individual can attempt a trip only a finite number of times, ultimately limited by its lifespan, which sets an upper bound on the ability of a single individual to improve its learned route over time.

One way in which animals can overcome the finite time horizon set by an individual lifespan is by genetically encoding migratory information—such as when and where to migrate—so that it can be inherited by future generations (*Liedvogel et al., 2011*; *Väli et al., 2018*). However, encoding information genetically can be slow and susceptible to environmental variations (*Cavalli-Sforza and Feldman, 1973*; *Whiten et al., 2017*; *Whitehead et al., 2019*). Alternatively, traveling in a group can allow less-informed individuals to incorporate information from others into their own learned route, thereby permitting information to pass across individuals (and time). When such social learning is coupled with the ability to recognize beneficial solutions and prune out costly mistakes, improved solutions can be preferentially spread horizontally and/or vertically. Such 'ratchet'-like improvements are referred to as cumulative cultural evolution (CCE), in recognition of its functional similarity to genetic evolution (i.e., variation, heritability, and selection). CCE has been observed in humans and some non-human animals

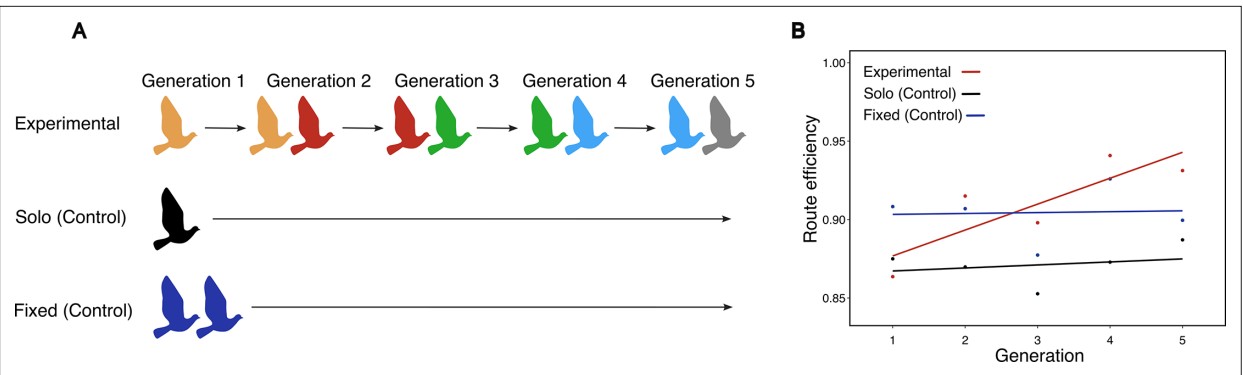

**Figure 1.** Schematic illustration of the experimental design used in the original study. (**A**) In each generation, birds were flown either individually or in pairs for a total of 12 releases. In the experimental condition, partners were replaced in each generation (except the first), whereas in the control conditions, birds were flown either individually or as fixed pairs for a total of 60 releases. (**B**) Variation in route efficiency in each condition across generations. Route efficiency was calculated from the final flight of each generation (release 12) for the experimental condition, and from releases 12, 24, 36, 48, and 60 for the solo and fixed controls. Panel A and B are adapted from Figure 1 and 2 from *Sasaki and Biro, 2017*.

(*Boyd and Richerson, 1996*; *Tomasello, 2009*; *Dean et al., 2014*; *Creanza et al., 2017*; *Miton and Charbonneau, 2018*; *Whitehead et al., 2019*), including primates (*Yamamoto et al., 2013*; *Whiten et al., 2022b*; *van Leeuwen et al., 2024*), birds (*Hunt and Gray, 2003*; *Sasaki and Biro, 2017*; *Aplin, 2019*; *Williams and Lachlan, 2022*), and cetaceans (*Krützen et al., 2005*; *Mann et al., 2012*; *Garland et al., 2022*). CCE is usually inferred as the underlying process when improvements 'accumulate' over time, even though the social learning mechanism(s) that lead to these improvements have not been directly observed (see *Mesoudi and Thornton, 2018* for core criteria required for CCE and *Hunt and Gray, 2003*; *Claidière et al., 2014* for examples of such cases). Yet, as previously discussed, alternative mechanisms such as the wisdom of crowds could in principle also cause an increase in task performance, without the need for other components associated with CCE (*Mesoudi and Thornton, 2018*).

In this study, we build upon one such example, by *Sasaki and Biro, 2017*, in which homing pigeons (*Columba livia*) were shown to improve their navigational performance over time, but the mechanism by which they do so remains a mystery. Homing pigeons are a highly social species that have been extensively studied for their ability to develop idiosyncratic routes and recall them across repeated flights, making them an ideal model organism for studying navigational strategies (*Matthews, 1951*; *Wallraff, 2001*; *Nagy et al., 2010*; *Flack and Biro, 2013*; *Nagy et al., 2013*; *Sasaki and Biro, 2017*). In their experiment, Sasaki and Biro created 'chains' of birds, similar to a game of telephone, and allowed them to fly back home repeatedly from a release site 8.4 km away. In the experimental condition, each chain was composed of five 'generations', with each generation consisting of 12 flights home. In each generation, an experienced bird, having developed knowledge about the homing task from the previous generation, was paired with a naïve bird that lacked this information. At the end of the generation, the experienced bird was removed and a new naïve bird was paired with the remaining (now-experienced) bird (see 'Methods' for details). As control conditions, solo and fixed pairs of birds were flown for the same total amount of flights as in the experimental condition (60 flights) (*Figure 1A*). To quantify the performance of the birds, the route efficiency, calculated as the ratio of the beeline distance (straight line distance between the release and home sites) to the actual distance traveled by the pair, was measured (a value of 1 indicates a beeline flight, while a value of 0 indicates an infinitely long flight).

The study found that the experimental chains of birds significantly outperformed both the solo and fixed pair controls by the end of the fifth generation (*Figure 1B*). The researchers proposed CCE as the mechanism driving this result, where chains of birds improve their routes by exploring different options and selectively choosing better ones. Valentini et al. examined the trade-offs between exploration and exploitation of new routes by the naïve and experienced birds (*Valentini et al., 2021*), showing that both birds had an equally likely chance of initiating collective explorations. Despite these insights, a detailed mechanistic understanding of how route improvements emerge in pigeon chains is still lacking. For instance, it remains unclear what (social and/or environmental) information birds

are sensing and using, and what kinds of computations they are performing on that information to produce the observed increase in route efficiency across generations.

Here, we hypothesized several plausible learning mechanisms that bird pairs might employ during homing navigation. We simulated each of these mechanisms using artificial chains of birds composed of trajectories from the solo control dataset and applied a weighted averaging procedure (which depended on the mechanism under study) to mimic the outcome of paired flights in a generation, analogous to the actual experimental design. It is also important to clarify that we use the terms *mechanism* or *strategy* in a more abstract manner rather than in a purely mechanistic way. In other words, we model how two trajectories (representing the 'preferred' path of two birds) map to a new trajectory (representing the joint path of the two birds flying as a pair), rather than the moment-to-moment decisions of each bird. This is because neither the underlying biological basis of the birds' decisions, nor different sources of noise (e.g., perceptual noise, decision noise, environmental noise) can be uniquely inferred using the available geospatial data. Moving forward, we will be using the terms mechanism(s) or strateg(ies) interchangeably to indicate these meta-mechanism(s), while the actual cognitive mechanisms that lead to these meta-mechanisms arising remain an open question. Our learning strategies can be organized into three classes based on the kind of information required to execute the strategy. We quantitatively compared the predictions generated by each strategy with the experimental data to infer which strateg(ies) real birds are most likely to be using. By doing so, we can assess what kind of information and cognition pigeons may be using during paired homing flights. In particular, we ask whether the cognitive requirements of CCE are necessary for the observed improvement in navigation ability.

## The navigational model

To simulate the experimental condition, we created artificial chains of bird pairs—similar to the experimental design—using bird trajectories from the solo control condition. We used the 12th flight of each solo control bird, since birds typically established stable idiosyncratic routes by this point, with minimal improvements in subsequent releases. In our model, we assumed that when two birds are paired together, a shared route is formed from their individual routes, which is the outcome of the learning process that the two birds engage in over the course of 12 releases. We model this learning process using a weighted averaging procedure for each pair (see 'Methods' for details). The naïve bird in that pair replaces its idiosyncratic route with this new averaged trajectory and becomes the 'experienced' bird in the next generation, where it is paired with a new naïve bird drawn from the solo control condition to continue the sequence. This iterative process—sampling a bird, generating the averaged route, assigning it as the 'experienced' bird for the next generation, and pairing it with another bird—was repeated for five generations, matching the actual experimental chains. The weighting parameter $w_E$ used in our model quantifies the influence (i.e., leadership) of the 'experienced' bird. As a comparison, for the experimental birds, we measured leadership based on the relative front-back positions of the birds during flights (since leaders have been found to take frontal positions across many species; see 'Methods' for details). We compared the route efficiencies and social weights of the experienced bird resulting from each simulated strategy with the experimental data to identify which candidate strateg(ies) are compatible with the empirical data.

We simulated seven strategies, which can be categorized into three types based on the kind of information required to execute the strategy. The first type ($T_1$) represents the simplest process, where birds have no knowledge of their partner's level of experience or performance (*Figure 2A*). This includes only the *averaging* strategy, where both birds weigh their individual routes equally in each generation, regardless of their respective navigational abilities.

The second type ($T_2$) assumes that birds are capable of recognizing the more experienced individual within the pair and adjust their social weighting to maximize performance using this knowledge (*Figure 2B*). Indeed, empirical evidence suggests that individuals can rely less on social information as they gain more experience (*Brønnvik et al., 2024*). Within $T_2$, the *experienced bird weighting* strategy assumes that the experienced bird maintains a constant weight $w_E$ across all generations, where $w_E$ is chosen such that route efficiency, on average, is maximized for the fifth generation. Alternatively, the *maximize generation* strategy allows the experienced bird to adjust its weight $w_E$ based on the generation the bird is in, thereby maximizing the route efficiency for each generation.

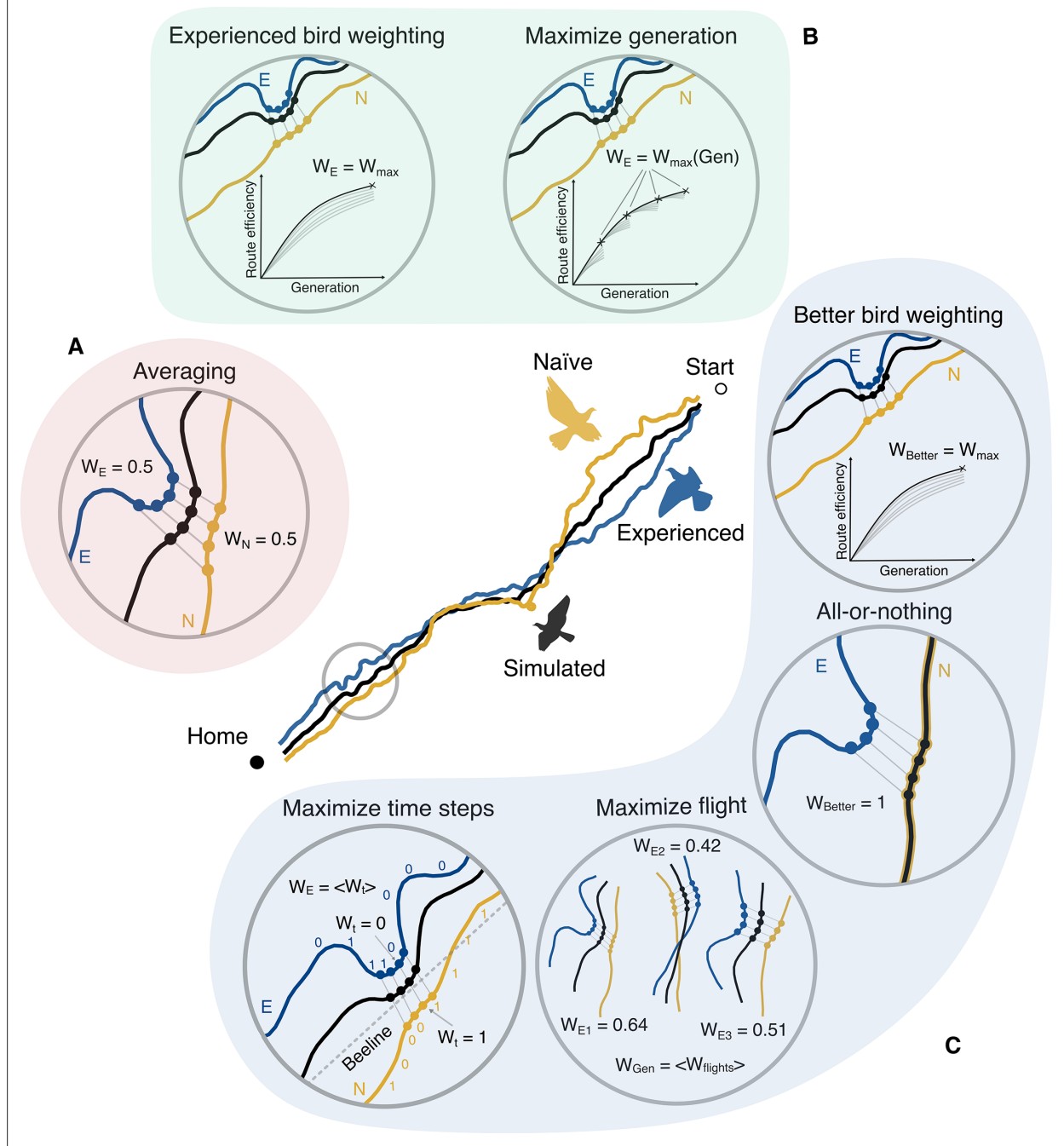

**Figure 2.** Illustration of the hypothesized social learning strategies. At the center, we depict an example of a simulated averaged bird trajectory (which becomes the experienced bird's route for the next generation) created using an experienced bird (E) and a naïve bird (N) trajectory. The strategies can be organized into three types based on the kind of information that is required to execute the strategy: (**A**) $T_1$ (red; requiring no information)—the simple process of equally *averaging* the two bird routes. (**B**) $T_2$ (green; requiring individuals to recognize relative experience levels)—strategies that optimize the weight of the experienced bird either to maximize performance in the final generation (*experienced bird weighting*) or in each generation (*maximize generation*). (**C**) $T_3$ (blue; requiring individuals to compare the relative path length of flights)—strategies where birds assess paired performance, either by assigning greater social weight to the better-performing bird (*better bird weighting* and *all-or-nothing*) or by fine-tuning local aspects of the flight (*maximize flights* and *maximize time steps*).

The third type ($T_3$) introduces the highest level of cognitive complexity, requiring birds to actively evaluate their individual or paired performance (*Figure 2C*). The strategies in $T_3$ align with the mechanistic criteria required for CCE, where individuals can recognize and select for better-performing routes. Within $T_3$, the first strategy, *better bird weighting*, assigns the better bird (i.e., the bird in the pair that has a shorter individual route, regardless of its experience level) a weight $w_B$ that maximizes efficiency in the fifth generation. Here, information from the worse-performing bird is still incorporated in the weighting process (with a weight $1 - w_B$) before passing to later generations. A more extreme version of this strategy is the *all-or-nothing* strategy, wherein the better bird takes full despotic control (i.e., $w_B = 1$), propagating only the best-performing routes across generations, with no contribution from the worse bird. The remaining two strategies focus on adjusting 'local' aspects of a paired flight. In the *maximize flight* strategy, birds aim for finding the best achievable route as a pair. For a particular pair of birds, this is a unique pair of weights $w$ and $1 - w$ that maximizes the route efficiency of the resulting trajectory. Finally, the *maximize time steps* strategy assumes that birds can judge their joint performance moment-to-moment and make adjustments to their heading at each time step during a flight. In this strategy, the bird with a preferred route closer to the beeline at a given moment, on average, exerts greater influence over the paired flight, pulling the pair closer to the correct homing direction.

Collectively, these strategies range from the minimal cognitive requirement ($T_1$) to recognizing one's own experience level ($T_2$) to evaluating the relative quality of the preferred routes of the two birds ($T_3$). We systematically tested the seven strategies by generating all possible combinations of chains from the solo control trajectories and measured the resulting distribution of route efficiencies and social weights used by the experienced bird for each generation. Using a mixed modeling approach, we fitted two different models, one for the responses of route efficiency and one for the social influence of the experienced bird (see 'Methods' for details) and compared the results to the experimental data.

## Results
### Similar route improvements emerge from simple and complex strategies

Across all strategies, we observed an improvement in route efficiency over generations (*Figure 3*). This suggests that a wide range of decision-making mechanisms can lead to navigational improvements,

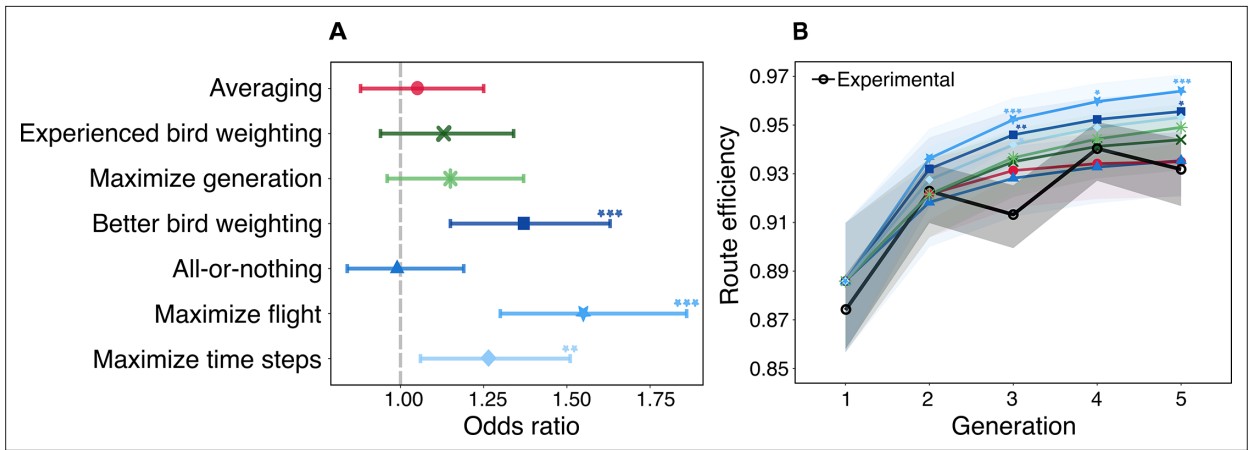

**Figure 3.** Results of mixed model analysis comparing the route efficiency of the hypothesized strategies to the experimental data. (**A**) The odds ratio for each strategy relative to the experimental data after adjusting for generation, with the gray dotted line representing the null odds ratio of 1. (**B**) The route efficiency of each hypothesized strategy and the experimental data (black) across generations. The shaded confidence bands indicate the 95% confidence intervals for the model predictions and reflect the biological uncertainty in the empirical dataset, not simulation noise. Statistical significance levels are denoted as follows: *p<0.05, **p<0.01, ***p<0.001. The general color scheme in the plots matches that used in *Figure 2* to highlight different strategy types.

not just the ones compatible with the definition of CCE. Interestingly, we found that the experimental birds performed relatively poorly compared to the majority of our hypothesized strategies (*Figure 3B*).

After adjusting for generation effects, we found that three strategies significantly outperformed the experimental birds in efficiency gains (*Figure 3A* and *Appendix 1—table 5*). These strategies all belonged to $T_3$ (where birds are able to evaluate route performance) and comprised the *better bird weighting* (OR = 1.37, 95% CI = [1.15, 1.63], p<0.001), the *maximize flight* (OR = 1.55, 95% CI = [1.30, 1.86], p<0.001), and the *maximize time steps* (OR = 1.26, 95% CI = [1.06, 1.51], p<0.01) strategies, with the *maximize flight* strategy (where a pair of birds uses the specific set of social weights that optimizes their joint route) producing the highest route efficiency among all strategies tested. These observed effects suggest that the ability to evaluate the performance of routes ($T_3$) can result in greater improvements compared to strategies that do not measure route efficiency ($T_1$ and $T_2$).

Furthermore, we found that both strategies within $T_2$ (where birds are able to detect which is the more experienced one), the *experienced bird weighting* (OR = 1.13, 95% CI = [0.94, 1.34], p=0.34) and the *maximize generation* (OR = 1.15, 95% CI = [0.96, 1.37], p=0.19) strategies showed some effect but had non-significant efficiency gains compared to the experimental data.

The remaining two strategies—*averaging* (OR = 1.05, 95% CI = [0.88, 1.25], p=0.93) ($T_1$) and *all-or-nothing* (OR = 0.99, 95% CI = [1.15, 1.63], p=1.00) ($T_3$)—aligned closely with the experimental data with little to no difference in improvements compared to the experimental birds. Interestingly, the *all-or-nothing* strategy, which passes the best routes without integrating any information from the worse-performing bird (i.e., a greedy strategy), performed almost identically (poorly) to the simplest strategy of *averaging* individual routes. This is despite the fact that the *all-or-nothing* strategy has the ability to identify which of the two birds knows the shorter route, while the *averaging* strategy does not.

Examining how the performance of these strategies varied across generations provides further insights into the dynamics of route improvements (*Figure 3B* and *Appendix 1—table 7*). Several strategies showed significant positive effects, but only from generation 3 onwards. These included the *better bird weighting* (Gen 3: OR = 1.66, 95% CI = [1.16, 2.39], p<0.01; Gen 5: OR = 1.57, 95% CI = [1.05, 2.37], p=0.02), the *maximize flight* (Gen 3: OR = 1.89, 95% CI = [1.32, 2.72], p<0.001; Gen 4: OR = 1.51, 95% CI = [1.00, 2.27], p=0.046; Gen 5: OR = 1.96, 95% CI = [1.30, 2.94], p<0.001), and the *maximize time steps* (Gen 3: OR = 1.54, 95% CI = [1.07, 2.22], p=0.01; Gen 5: OR = 1.49, 95% CI = [0.99, 2.24], p=0.06) strategies. However, the increase in effect in generation 3 specifically can be attributed to the dip in efficiency of the experimental birds during this generation, inflating the contrasts with other strategies.

Overall, these observations helped us narrow down our search for candidate mechanisms, excluding three strategies from $T_3$ which showed significant deviation from the experimental data. However, the strategies in $T_1$ and $T_2$, as well as the *all-or-nothing* strategy in $T_3$, remain statistically indistinguishable from the experimental data when comparing their route efficiencies.

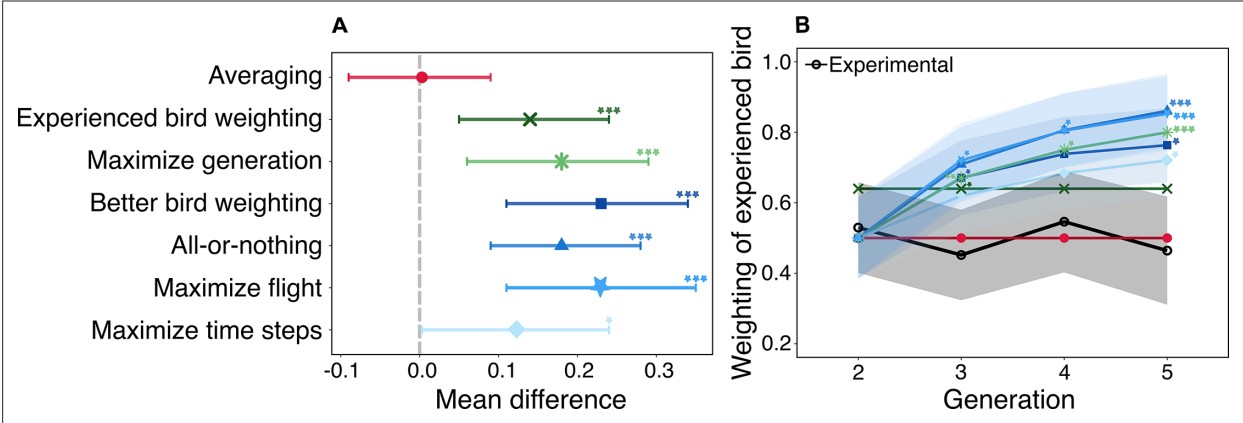

**Figure 4.** Results of mixed model analysis for the social weighting of the experienced bird. (**A**) The mean difference for each strategy relative to the experimental data after adjusting for generation, with the gray dotted line representing the mean difference of 0 between the datasets. (**B**) The variation of the social weighting assigned to the experienced bird for each strategy across generations. The shaded confidence bands indicate the 95% confidence intervals for the model predictions and reflect the biological uncertainty in the empirical dataset, not simulation noise. Statistical significance levels are denoted as follows: *p<0.05, **p<0.01, ***p<0.001. The color scheme in the plots is the same as in *Figure 2*.

## Experimental birds rely on simple route averaging for navigation

Next, we examined how the social weights of the birds varied across different strategies in comparison to the experimental birds (*Figure 4*). Although several of our hypothesized strategies assigned social weights to the better/worse bird ($T_3$), rather than the experienced/naïve bird ($T_1$ and $T_2$), for all strategies we calculated the mean social weight of the experienced/naïve bird to facilitate comparison with the experimental data. Furthermore, because the social weights of a pair of birds sum to 1, we report just the social weight of the experienced bird in this analysis.

Certain strategies—*averaging*, *experienced bird weighting*, and *maximize generation*—had, by definition, constant social weights assigned to the experienced bird. Specifically, the *averaging* strategy assigned a weight of 0.5, meaning that both birds contributed equally to homing decisions. For the *experienced bird weighting* strategy, the social weights were optimized to achieve maximum efficiency in the last generation, which was estimated to be 0.64 for the experienced bird. For the *maximize generation* strategy, the weights were optimized for each generation, with the experienced bird assuming weights 0.5 (generation 2), 0.67 (generation 3), 0.75 (generation 4), and 0.80 (generation 5) (note that this appears to follow a proportional relation $1/M$ for $M$ generations, which we will return to later). Given their fixed weights, we excluded these strategies from our mixed model and manually contrasted their effects with the experimental birds.

After adjusting for generation effects, we found that the leadership values of the experienced bird in the experiment aligned with only one of the strategies: *averaging* (mean difference = 0.003, 95% CI = [–0.08, 0.09], p=0.99) with almost no difference between the two (*Figure 4A* and *Appendix 1—table 6*). In other words, the experienced bird and the naïve bird shared, on average, equal social weight across all generations. All other strategies predicted a significantly higher influence of the experienced bird compared to that observed in the experimental data (*experienced bird weighting* [mean difference = 0.14, 95% CI = [0.05, 0.23], p<0.001], *maximize generation* [mean difference = 0.18, 95% CI = [0.09, 0.27], p<0.001], *better bird weighting* [mean difference = 0.17, 95% CI = [0.06, 0.29], p<0.001], *all-or-nothing* [mean difference = 0.23, 95% CI = [0.11, 0.34], p<0.001], *maximize flight* [mean difference = 0.23, 95% CI = [0.11, 0.34], p<0.001], and *maximize time steps* [mean difference = 0.12, 95% CI = [0.00, 0.23], p=0.04]). Combined with our previous conclusions, only one of our hypothesized strategies quantitatively matched both the route efficiency of the real birds (for all generations) as well as the social weights of the real birds (for all generations): the *averaging* strategy.

We observed that the average influence of the experienced bird monotonically increased across generations, except for the *averaging* and *experienced bird weighting* strategies (*Figure 4B* and *Appendix 1—table 8*). By the end of generation 5, the strategies that showed the strongest positive deviations from the experimental birds included the *better bird weighting* (mean difference = 0.30, 95% CI = [0.08, 0.52], p<0.01), *all-or-nothing* (mean difference = 0.40, 95% CI = [0.18, 0.62], p<0.001), *maximize generation* (mean difference = 0.34, 95% CI = [0.12, 0.56], p<0.001), *maximize flight* (mean difference = 0.39, 95% CI = [0.17, 0.61], p<0.001), and *maximize time steps* (mean difference = 0.26, 95% CI = [0.04, 0.48], p=0.01).

Interestingly, we found that strategies that had similar average social weights can lead to very different route efficiency outcomes. For instance, the *all-or-nothing* and *maximize flight* strategies had almost identical trends for the social weights of the experienced bird. However, these strategies had substantially different route efficiency outcomes, with the *all-or-nothing* strategy being the worst performing and the *maximize flight* strategy achieving the highest efficiency gain. Conversely, strategies that exhibited similar route efficiencies could be driven by very different average social weights. For example, the *all-or-nothing* and *averaging* strategies produced similar route efficiency outcomes, yet the former had the highest influence of the experienced bird, while the latter had the lowest influence. These observations demonstrate that the average social weights are not predictive of performance, nor is performance predictive of social weights, at least in their aggregate.

## Chaining with replacement experimental design leads to reduced effective group sizes

Although homing pigeons appear to be simply averaging their flights when flown together, the structure of the 'chaining with replacement' experimental design does not result in the five birds in a chain equally contributing to the final learned flight. Instead, the chain structure produces a 'geometric forgetting' of past information, with the last bird contributing 1/2 of the final flight path,

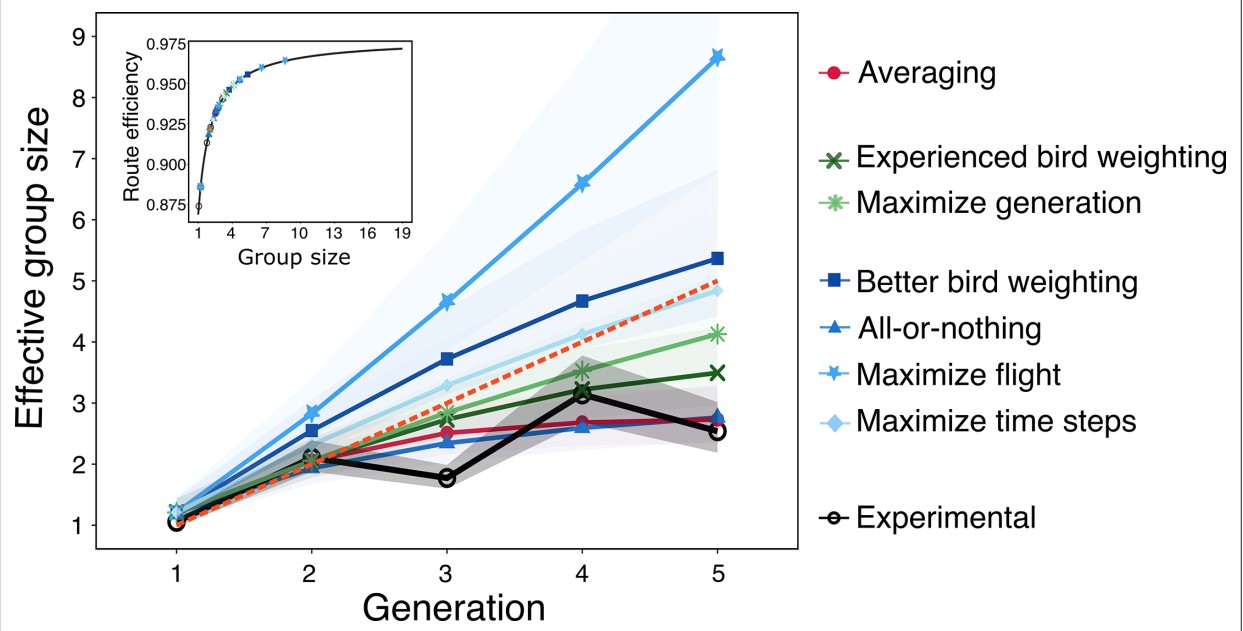

**Figure 5.** The effective group size achieved by each strategy and the experimental data across generations. The predicted mean route efficiency from each strategy is mapped onto a curve modeling the relationship between the route mean route efficiency and simulated flock sizes. The inset provides the curve, illustrating how route efficiency scales with flock size. The dashed orange line shows the 1:1 between the two variables. Shaded confidence bands denote the 95% confidence intervals for the estimated effective group sizes.

the penultimate bird contributing a 1/4, the third bird contributing 1/8, etc. Indeed, we predict that the first bird contributes only 1/32 of the final flight path. These unequal contributions of each bird to the final flight trajectory are in contrast to the classic 'wisdom of crowds', for which collective performance is maximized when all individuals are weighed equally (assuming that individuals are unable to perceive differences in competence).

We asked to what extent the chain structure, and correspondingly the unequal weighting of birds, decreases the collective performance of the group. To answer this, we defined a measure that we call 'effective group size', which is the size of a democratic (i.e., equally weighted) group that exhibits equivalent performance to the focal group under study (which is composed of a certain number of individuals, each with a certain social weight). To estimate this, we first mapped the size of democratic flocks to a corresponding route efficiency (*Figure 5*, inset) by simulating the outcome of democratic flocks through averaging the routes of independent control flights and measuring the route efficiency of the resulting collective flights (see 'Methods' for details). We then re-mapped the route efficiencies resulting from our hypothesized strategies, as well as the experimental data (i.e, *Figure 3B*), to an equivalent effective group size.

Most of the hypothesized strategies exhibit an effective group size smaller than five by the end of the fifth generation (i.e., fall below the 1:1 line shown as a dashed orange line in *Figure 5*). This demonstrates that the chain structure tends to be an inefficient way to combine information, since for these strategies, the five birds that comprise a chain show a diminished performance compared to a hypothetical flock of five birds flying together. This is true even for all of the strategies of $T_2$, where there is knowledge of who is the more experienced bird (knowledge that our hypothetical flock of five birds does not possess). By contrast, two of the strategies of $T_3$ exhibit an effective group size greater than five. This is possible because birds using this strategy have direct knowledge of the quality of a route, information that is unavailable to our simulated flocks of five birds.

Our results suggest that the experimental design limits the propagation of information from the past to the future. For a chain of five birds, the average effective group size from the experimental dataset barely exceeds two birds (2.12) (*Figure 5*). This suggests that in each generation, contributions from prior individuals are hardly retained and chains of experimental birds fly only marginally better than pairs of birds. In contrast, more complex navigational mechanisms, facilitated by, for example, perceiving route quality, could surpass this limit and even reach effective group sizes exceeding five

individuals. Hence, continuous replacement of birds within a chain leads to a substantial loss of information, and, unless compensated by a more cognitively complex mechanism, constrains the chain's ability to improve the efficiency of its route over time.

## Discussion

In this study, we investigated potential navigational strategies, which varied in the level of cognitive complexity required to execute them, to infer the mechanism of collective route improvements in homing pigeons. Our work builds upon a 2017 experiment which demonstrated that pairs of experienced and naïve pigeons could iteratively improve homing routes over generations. Specifically, our aims were threefold: (1) provide insights into the mechanism(s) pigeons use to improve route efficiency, (2) investigate how cognitive complexity, as well as experimental design (turnover of birds), relate to performance, and (3) investigate if the identified mechanism(s) fall under the criteria required for CCE. To achieve this, we developed seven plausible mechanisms, broadly categorized into three types ($T_1$, $T_2$, and $T_3$), with increasing cognitive complexity. We created a simple navigational model by artificially pairing solo bird trajectories, similar to the experimental design, and tested our strategies to identify candidate strateg(ies) that are quantitatively compatible with the empirical data. All of our strategies resulted in route improvements, regardless of their underlying complexity. However, when combined with the results from the social weight analysis, the experimental data overall aligned best with the simplest strategy: *averaging* individual routes.

The route efficiency achieved generally increased with the cognitive complexity of the strategy—strategies in $T_3$ tended to perform the best. However, this was not always the case. Specifically, the *all-or-nothing* strategy ($T_3$), where only the best route was passed to the next generation, performed no better than the simple *averaging* strategy ($T_1$). This can be explained by recalling that there are two sources of information: leadership from experts and emergent wisdom of crowds. The *all-or-nothing* and the *averaging* strategies, respectively, represent these two 'pure' strategies. By contrast, all of the other strategies represent mixed strategies, whereby individuals with better information have greater social weight, but individuals with worse information still contribute to the collective decision. The mixed strategies all outperformed the pure strategies, demonstrating the benefit of exploiting both sources of information.

Furthermore, the strategies in $T_3$ are compatible with the definition of CCE (*Tomasello, 2009*; *Mesoudi and Thornton, 2018*) but showed a quantitative mismatch with the experimental data. Therefore, an improvement in route efficiency alone is not sufficient evidence for cultural transmission, as the experimental birds did not exhibit evidence of some of the criteria of CCE (evaluation of performance, for instance). This is because of the possibility of the wisdom of crowds improving routes 'for free' without the need for individuals to specifically recognize and maintain improvements. In other words, the wisdom of crowds functions like a statistical ratchet, rather than a deterministic one. Moreover, as evidenced from our social weighting analysis, we did not find direct evidence of social learning in terms of imitative copying of prior routes (see *Pettit et al., 2013* for similar results). Birds in this experiment, on average, equally influenced each other's routes disregarding any differences in experience. This observation raises broader questions about which social learning mechanisms truly align with the requirements for CCE (*Whiten et al., 2022a*; *Dalmaijer, 2024*).

We note that the wisdom of crowds is beneficial in some but not all task types (*Anderson et al., 2001*; *Sasaki et al., 2022*; *Langridge et al., 2008*). In the context of the homing task, the pigeons apparently have a general (noisy) tendency to find their way home. Because of this, averaging the flights of multiple pigeons is an effective way of canceling out noise, resulting in an increasingly efficient route (on average). Hence, averaging routes can lead birds to stumble upon the best routes by chance. By contrast, in human cultural evolution, complex innovations are unlikely to arise *de novo* by averaging existing behaviors (*Creanza et al., 2017*; *Migliano and Vinicius, 2022*). In these types of tasks, selection on novel behaviors at the tails of the distribution—not the average—is probably necessary to 'evolve' towards improved solutions. Future work should investigate the range of tasks that group-living animals typically face, and categorize them into ones that can benefit from the wisdom of crowds and ones that require CCE (*Gruber et al., 2022*).

This study also reveals that collective strategies must be understood within the context in which they are implemented. Specifically, here the experimental design caused birds to fly in pairs in a particular chain structure. Combined with the *averaging* strategy, this resulted in a geometric dilution

of influence over generations. This progressive loss of information due to repeated turnover of birds reflects an inefficient version of the 'wisdom of crowds', where instead of equally incorporating the knowledge of all past individuals, information from early generations is disproportionately lost. This is further supported by our effective group size analysis, which we developed as a reference metric to quantify the effective number of individuals contributing towards route improvements in each generation. We find this to be a potentially valuable metric to quantify such chained experimental design, which are usually proposed for studying CCE (*Caldwell and Millen, 2008*). Using this method, we showed that despite consisting of five birds, experimental chains of birds exhibited an effective group size only slightly better than two individuals. We expect the behavior of animals to be adaptive in the conditions in which they evolved. The chain structure of this experiment may approximate the dynamics of fission-fusion groups, and additionally forced group sizes to be just two birds. While urban pigeon flocks are highly fluid, they are typically of sizes much larger than two (*Lefebvre, 1985*). The *averaging* strategy that the experimental birds exhibited may have performed relatively poorly within this chain structure, but it should exhibit higher performance in larger groups, where there is more scope for the wisdom of crowds. A simple strategy such as the *averaging* strategy may also be beneficial in large temporary groups, where recognizing experienced or more knowledgeable individuals may be difficult.

It is also important to consider what the pigeons themselves are trying to optimize (*Gruber et al., 2022*). While our analysis focused on route efficiency, pigeons may not necessarily prioritize minimizing travel distance. Indeed, the average difference in travel distance between the best-performing and worst-performing strategies is only a few hundred meters. Previous studies have shown that pigeons rely on specific landmarks for navigation, suggesting that familiar, slightly longer routes may be preferable to unfamiliar, shorter ones (*Meade et al., 2005*; *Guilford and Biro, 2014*; *Biro et al., 2004*). In this context, birds may be prioritizing maximizing their probability of successfully returning home rather than minimizing flight distance per se (which would require exploring alternative routes, with an associated risk of getting lost). The *averaging* strategy may be best for birds to maximize the probability of returning home.

Another assumption of our model is that the birds execute the strategies perfectly. In other words, the birds are able to accurately identify who is the more experienced bird or the bird with the shorter route, and can apply the optimal social weight for that strategy. Real animals, if they did implement these strategies, would have some error rate (either in differentiating the birds or applying the optimal social weight), resulting in a lower measured performance. If the error rate is high enough, the performance could even be worse than that exhibited by the *averaging* strategy. Therefore, the *averaging* strategy could be a less risky one, guaranteeing (statistically) some measure of improvement.

Our results highlight the need for a more general framework for understanding CCE, calling for a critical re-evaluation of its definition and evaluative criteria, especially in the context of non-human animals. Innovations can be the emergent outcome of collective behavior, rather than necessarily originating from a single individual (*Berdahl et al., 2013*). In that case, these collectively generated innovations can transmit within the group, without necessarily relying on social learning or leadership. When individuals lack the ability to recognize beneficial innovations, there is no ratchet in a strict sense, but a ratchet may nonetheless exist statistically, due to the wisdom of crowds. Moreover, CCE and the wisdom of crowds are not necessarily mutually exclusive, and social animals could, in principle, simultaneously exploit both sources of innovation, as illustrated in the mixed strategies we explored. We therefore recommend that future empirical studies of CCE in non-human animals clearly distinguish between these mechanisms and incorporate additional criteria for evaluation. Marrying the fields of collective intelligence and CCE could provide a powerful framework for understanding how social animals can improve their fitness across a range of task types and cognitive abilities.

In conclusion, our study opens avenues for future research into collective decision making across time (*Biro et al., 2016*). Collective performance is the emergent outcome of the social structure, task type, and social strategy. Understanding how these interact with each other could improve our predictions about what collective strategies we should expect to observe in nature given uncertain shifts in timings of events in the future (*Kao et al., 2024*). For this navigation task, simple averaging is sufficient to quantitatively explain the experimental results, but other tasks may be less amenable to the wisdom of crowds. It is crucial to understand collective strategies within the context in which they evolved, including typical group sizes, turnover rate, error rate, and how many times a task is

repeated. While more cognitively complex strategies can result in greater performance, they may not be necessary, and, indeed, mixed strategies that draw on both individual and collective sources of intelligence may result in the highest levels of performance.

## Methods
### Data pre-processing and experimental procedure

We used GPS trajectory data from a previous study by *Valentini et al., 2021*, which was based on a 2017 study conducted by Sasaki and Biro in homing pigeons (*C. livia*) (*Sasaki and Biro, 2017*). The study consisted of three conditions: experimental, solo control, and fixed control. Each condition consisted of birds that were allowed to fly back home from a site 8.4 km away, either individually or as pairs. The experimental trials were structured around sets of 12 bird flights, termed a 'generation'. In generation 1 of the experimental group, a single bird flew 12 times before being paired with a new individual (the naïve bird) in generation 2. This pattern continued in subsequent generations, wherein the prior naïve bird (now experienced) was paired with a new bird, creating a transmission chain. Each chain consisted of five generations, or 60 total flights across the entire chain. The birds in the control conditions completed an equal number of flights (60 releases), with solo control birds flying individually and the fixed control birds flying in constant pairings. In total, each condition had 10 independent replicate chains.

For our analysis, we focused on the final flights (release 12) of each generation for the experimental chains for comparison with our model predictions. Following Valentini et al., we only considered flights where the two birds were less than 250 m from each other at all times, as well as excluding flights of birds that never reached their home destination (see 'Materials and methods' in *Valentini et al., 2021*). After accounting for missing flights in the experimental group, we were left with 10 flights each for generations 1–3, 8 flights for generation 4, and 7 for generation 5. Each of the trajectories was down-sampled to a frequency of 1 Hz (from 5 Hz) using the R package `trajr` (*McLean and Skowron Volponi, 2018*). The trajectory points were converted from their original geographic coordinate system (GPS) to the Universal Transverse Mercator (UTM) system for all of our distance calculations. The joint route flown by a pair of birds was calculated as the center of mass of the two birds at each time point. The route efficiency was then obtained as the ratio of the beeline distance to the actual distance traveled by the birds as a pair. To avoid biases due to circling behavior of the pigeons near release and home sites, we excluded trajectory points within a radius of 800 m and 200 m from the release and home site, respectively. Our beeline distance was adjusted accordingly to account for these zones.

### Weighted averaging

To simulate the experimental chains of birds, we first assumed that each naïve bird has an initial 'preferred' route that it would follow if flown alone. We compiled a set of these preferred routes by taking the 12th release for each of the solo control birds, to match our procedure of analyzing the 12th release of each generation in the experimental condition. Moreover, the route efficiency of the solo control birds reached their maximum, on average, by this flight, and therefore represents the idiosyncratic learned preference of that bird.

We then used a weighted averaging method to simulate the consensus flight that would emerge when two birds are flown together. Each run in our simulation proceeds as follows: In generation 1, we randomly chose a trajectory from our set, which becomes the 'experienced' bird in generation 2. For generation 2, our experienced bird is paired up with a naïve bird, which is another trajectory sampled with replacement from the representative set. We then computed an averaged route from these paired flights using the following steps: First, we synchronized the number of data points across the two trajectories to match the size of the longer trajectory. This is done by interpolating missing coordinates from two consecutive time points in the shorter trajectory. We used the `TrajResampleTime()` function available from the R package `trajr` to perform our resampling (*McLean and Skowron Volponi, 2018*). Next, the x and y coordinates of the paired trajectories were combined using the following weighted averaging equations:

$$X_{\text{avg}}(t) = X_E(t) \cdot w + X_N(t) \cdot (1 - w),$$

$$Y_{\text{avg}}(t) = Y_E(t) \cdot w + Y_N(t) \cdot (1 - w),$$

where $w \in [0, 1]$ and $t \in \{t_1, t_2, \ldots, t_L\}$ correspond to the time steps along a trajectory of length $L$. Here, $w$ is the weighting parameter, which quantifies the social influence of the experienced bird ($E$) relative to the newly sampled naïve ($N$) bird. Therefore, a weighting of $w = 0$ indicates that the experienced bird has no influence on the averaged route, while $w = 1$ implies full control of the experienced bird on route formation. To account for all possible weighting contributions, $w$ was incremented in steps of 0.01 during simulations of each strategy. We assume that the previously naïve bird adopts this averaged route as its preferred route by the end of the generation, and this averaged route is then propagated as the experienced bird's preferred route for generation 3. A new naïve bird is drawn from our dataset, and its preferred route is again averaged with the experienced bird's. This process of sampling, pairing, averaging, and propagation continued iteratively using all possible combinations of birds in each generation, until generation 5. By systematically varying the weighting parameter $w$, we investigated seven plausible strategies of navigation and calculated the resulting route efficiencies, similar to the method described for the experimental group.

## Leadership calculation

Previous studies have shown that pigeons follow a front-back leadership, wherein the bird at the front of the flock assumes a greater leadership role compared to birds at the back (**Nagy et al., 2010**; **Nagy et al., 2013**). We calculated a leadership metric for the final release in each generation of the experimental data to compare with our simulation results. To achieve this, we used the method outlined in Valentini et al. (see 'Appendix 1, Methods' in **Valentini et al., 2021**) that is based on calculating the relative position of the birds from each other $d_{EN}$, at each time step and projecting it onto the direction of motion of the bird pair in the subsequent time step. We used the sign of $d_{EN}$ to determine which bird among the pair was leading at a given time step. The distance $d_{EN}$ is positive if the experienced bird is in front of the pair and negative if the naïve bird is leading the pair. For our analysis, we used the experienced bird as our reference bird in the pair. We calculated the average proportion of time spent by the experienced bird in the front for each flight and used this for comparing with the simulated social weightings.

## Statistical analysis

We utilized a mixed-effects modeling approach for comparing our simulation results with the experimental data. We created two different models to address our two response variables: route efficiency and social weight of the experienced bird across strategies. Since the original experiment provided data from only nine birds in the solo control condition, we were limited by the number of representative trajectories for our simulations. As a result, the same bird trajectories were reused in our simulated chains (i.e., we sampled with replacement), introducing an unwanted dependency in our analysis. This non-independence was evident across generations and simulated chains, effectively reducing the variance within each sample relative to the true variance. To account for this non-independence, we incorporated bird ID (1–9, corresponding to each solo trajectory) and its interaction with each generation as our random effect in our model. For the experimental data, we conditionally excluded the random effect term due to independent bird IDs in the experimental chains. Thus, the modeled bird-specific deviations in the random effect term $\gamma_{\text{Gen:bird}}$ are

$$\gamma_{\text{Gen:bird}} \sim \mathcal{N}(0, \sigma^2_{\text{dataset}})$$

where the random effect variance $\sigma^2_{\text{dataset}}$ is dataset specific (experimental vs. simulated).

To predict route efficiencies, we combined the experimental and simulated datasets and fitted a beta regression model with a logit link using the `glmmTMB()` function from the R package `glmmTMB` (see **Appendix 1—table 1** and **Appendix 1—table 2** for model details and performance checks) (**Brooks et al., 2017**). We used generation, strategy, and their interactions as fixed predictors for our models. Given this, our model equations for the expected route efficiency $\mu_{ij}$ for generation $i$ and strategy $j$ are

$$\text{logit}(\mu_{ij}) = \beta_0 + \beta_1 \text{Gen}_i + \beta_2 \text{Strategy}_j + \beta_3(\text{Gen}_i \times \text{Strategy}_j) + \gamma_{\text{Gen}_i:\text{bird}}$$

Similarly, for the response variable social weight of the experienced bird, we fitted a linear mixed effect model using the `lmer()` function from the R package `lme4` (see *Appendix 1—table 3* and *Appendix 1—table 4* for model details and performance checks; *Bates et al., 2015*). We used an identical fixed and random effect structure as the route efficiency model specified above. However, since the estimates for the averaging, better weighting, and maximize generation strategies were constant values, they were removed from the mixed model and contrasted manually with the experimental data. The model equation for the expected social weight $\omega_{ij}$ for generation $i$ and strategy $j$ is given by

$$\omega_{ij} = \beta_0 + \beta_1 \text{Gen}_i + \beta_2 \text{Strategy}_j + \beta_3 (\text{Gen}_i \times \text{Strategy}_j) + \gamma_{\text{Gen}_i:\text{bird}} + \epsilon_{ij}$$

From both our models, we computed estimated marginal means (EMMs) adjusting for the random effects, using the `emmeans` package in R (*Searle et al., 1980*). Post hoc analysis tests comparing the simulated and experimental data were also performed using the emmeans package. To validate our model predictions and results, we performed a bootstrapping procedure (see *Appendix 1—table 9* and *Appendix 1—table 10*). We generated bootstrap samples for both the experimental and simulated strategies for each generation, using a sample size matching the number of experimental samples in the respective generation, and conducted 10,000 runs. Our model predictions were in accordance with the bootstrap results within a reasonable margin of error. We adjusted the calculated p-values and confidence intervals for multiple comparisons within each generation using Dunnett's method against our experimental control.

## Effective group size analysis

We conducted an analysis to map the predicted route efficiencies from our model with those achievable by democratic flocks of varying group sizes (i.e., they make decisions by averaging the preferred routes of all of the individuals in the group). This approach enabled us to better understand the cumulative effects of different mechanisms within the 'chaining with replacement' experimental design used in the study. To achieve this, we first generated groups of different sizes by artificially pairing birds from the final flights of the experimental and solo control conditions in generation 1 (19 trajectories total). Using a weighted averaging approach, similar to our previous analysis (but with equal weights), we calculated an overall mean route from the individual trajectories. This mean route served as the measure of the flock's average route efficiency. We then calculated mean route efficiencies for each group size and observed a non-linear relationship between group size and route efficiency. To characterize this trend, we applied non-linear least squares regression (NLS) in R (*Bates and Watts, 1988*), using various functional forms with group size as the predictor and route efficiency as the response variable. Among these evaluated models, the best fit was obtained using a hyperbolic function (see *Appendix 1—figure 1* for further details regarding model comparisons and goodness of fit). Finally, we used the model coefficients to back-transform our simulated efficiencies and derive corresponding effective group size values.

## Acknowledgements

We would like to thank Almond Stöcker, Cooper Kimball-Rhines, and Jacob Adamczyk for their valuable feedback on our analysis. SCB, FAF, and ABK acknowledge support from the U.S. National Science Foundation (BRC-BIO DBI-2233416) and a generous anonymous gift fund.

## Additional information

### Funding

| Funder | Grant reference number | Author |
| --- | --- | --- |
| U.S. National Science Foundation | BRC-BIO DBI-2233416 | Albert B Kao |
| Anonymous gift fund | | Albert B Kao |

| Funder | Grant reference number | Author |
|--------|------------------------|--------|

The funders had no role in study design, data collection and interpretation, or the decision to submit the work for publication.

## Author contributions

Shoubhik Chandan Banerjee, Conceptualization, Formal analysis, Visualization, Writing – original draft, Writing – review and editing, Investigation, Methodology, Software, Validation; Fritz A Francisco, Formal analysis, Writing – review and editing, Methodology; Albert B Kao, Conceptualization, Writing – original draft, Writing – review and editing, Investigation, Methodology, Supervision

## Author ORCIDs

Shoubhik Chandan Banerjee ⓘ https://orcid.org/0000-0003-4533-2140
Fritz A Francisco ⓘ https://orcid.org/0000-0002-8258-2221
Albert B Kao ⓘ https://orcid.org/0000-0001-8232-8365

Reviewer #1 (Public review): https://doi.org/10.7554/eLife.108054.3.sa1
Reviewer #2 (Public review): https://doi.org/10.7554/eLife.108054.3.sa2
Author response https://doi.org/10.7554/eLife.108054.3.sa3

# Additional files

## Supplementary files

MDAR checklist

## Data availability

The dataset and code used for this study have been deposited in figshare with identifier "10.6084/m9.figshare.28950032" and can be accessed through this link: https://doi.org/10.6084/m9.figshare.28950032.v1.

The following dataset was generated:

| Author(s) | Year | Dataset title | Dataset URL | Database and Identifier |
|-----------|------|---------------|-------------|-------------------------|
| Banerjee S, Kao AB, Francisco F | 2025 | Dataset and supporting code for manuscript: "A tale of two birds: cognitive simplicity drives collective route improvements in homing pigeons | https://doi.org/10.6084/m9.figshare.28950032.v1 | figshare, 10.6084/m9.figshare.28950032 |

The following previously published dataset was used:

| Author(s) | Year | Dataset title | Dataset URL | Database and Identifier |
|-----------|------|---------------|-------------|-------------------------|
| Valentini G, Pavlic T, Walker SI, Pratt SC, Biro D, Sasaki T | 2021 | Data and code from: Naïve individuals promote collective exploration in homing pigeons | https://doi.org/10.6084/m9.figshare.14043362.v1 | figshare, 10.6084/m9.figshare.14043362 |

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

# Appendix 1

## Model evaluation

### Route efficiency model

We used `Gen` (generation), `strategy` (0 for experimental and 1–7 for simulated data), and `bird_ID` as our predictor variables with route efficiency as our response variable. Broadly we tested fitting different model formulations including random effects and variable relationships. The code for the models we formulated is available in the `pigeon_replace_model_performance.R` file. We used AIC to select the best-fitting model (*Appendix 1—table 1*).

**Appendix 1—table 1.** Route efficiency data model comparison.

| Model | df | AIC |
| --- | --- | --- |
| Linear Model | 41 | –2141657 |
| Linear Mixed Effect Model (random effect bird ID) | 42 | –2280167 |
| Linear Mixed Effect Model (random effect bird ID and Generation) | 42 | –2281106 |
| Beta Regression Model | 41 | –2388197 |
| Beta Regression Model (random effect bird ID) | 42 | –2526965 |
| Beta Regression Model (random effect bird ID and Generation) | 42 | –2527103 |

The lowest AIC value was achieved by our beta regression model (see 'Methods' in the main text for details). The model diagnostics are shown in *Appendix 1—table 2*.

**Appendix 1—table 2.** Model variance components and $R^2$ values.

| Metric | Value |
| --- | --- |
| Fixed Effect Variance | 0.0425 |
| Random Variance | 0.1080 |
| Residual Variance | 0.00048 |
| Marginal $R^2$ | 0.2815 |
| Conditional $R^2$ | 0.9968 |
| Intraclass Correlation Coefficient (ICC) | 0.7154 |

### Social weight model

We used similar predictor variables as the prior model and tested different model formulations, with social weight of the experienced bird as our response variable. The AIC results are shown in *Appendix 1—table 3*.

**Appendix 1—table 3.** Social weight model comparison.

| Model | df | AIC |
| --- | --- | --- |
| Linear Model | 21 | –18914.90 |
| Linear Mixed Effect Model (random effect bird ID) | 22 | –81533.43 |
| Linear Mixed Effect Model (random effect bird ID and Generation) | 22 | –81967.68 |

The best-fitting model was a mixed effect model with our random effect terms. Model diagnostics is shown in *Appendix 1—table 4*.

**Appendix 1—table 4.** Social weight model variance components and $R^2$ values.

| Metric | Value |
| --- | --- |
| Fixed Effect Variance | 0.0038 |

*Appendix 1—table 4 Continued on next page*

*Appendix 1—table 4 Continued*

| Metric | Value |
|---|---|
| Random Variance | 0.0257 |
| Residual Variance | 0.0429 |
| Marginal $R^2$ | 0.0527 |
| Conditional $R^2$ | 0.4075 |
| Intraclass Correlation Coefficient (ICC) | 0.3548 |

## Group size analysis

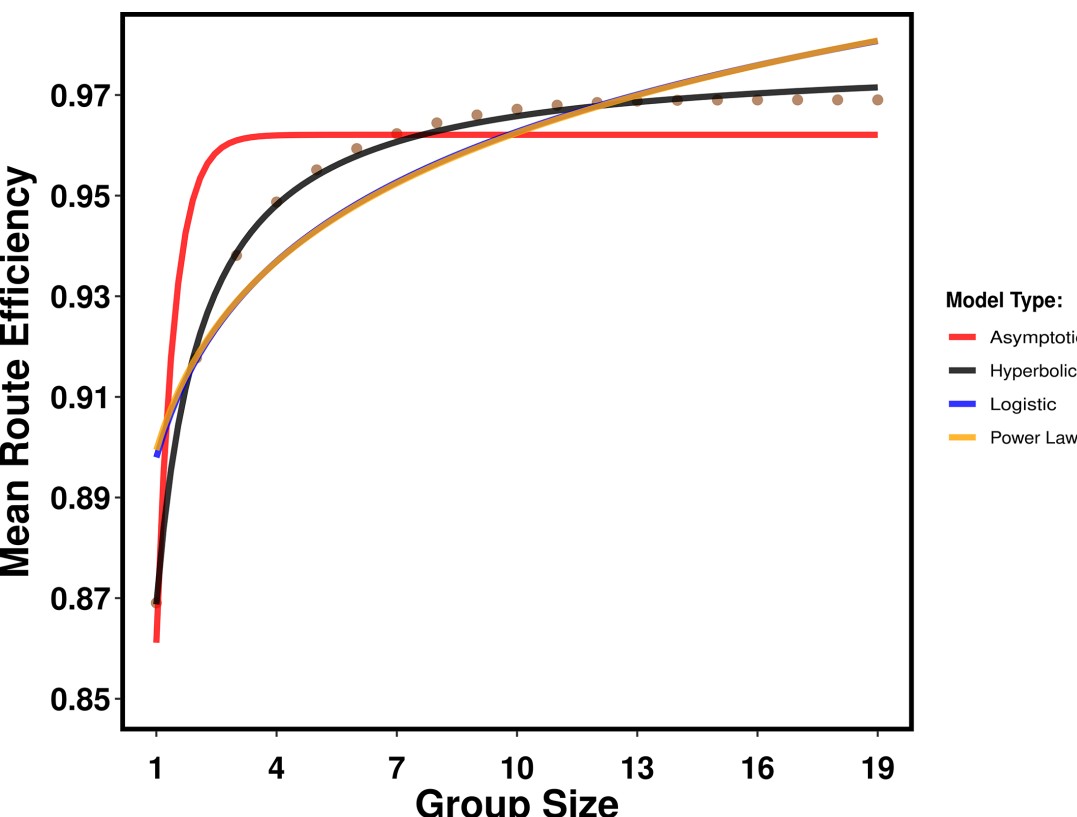

**Appendix 1—figure 1.** The figure illustrates various functional models fitted to the mean route efficiency values for a given group size. A post hoc model comparison was conducted using AIC to determine the best-fitting model: hyperbolic (df = 3, AIC = –189.8), logistic (df = 3, AIC = –114.6), power law (df = 3, AIC = –113.5), and asymptotic (df = 3, AIC = –110.5). Among these, the hyperbolic model provided the best fit. Note that the power law and logistic curves in the figure are nearly identical.

**Appendix 1—table 5.** Contrasts between estimated route efficiencies of each strategy and the experimental data holding generation constant (at its mean value).

Contrasts were conducted on the log odds ratio scale relative to the experimental data. Confidence intervals (CIs) were calculated at the 95% interval. Dunnett's correction was applied to both CIs and p-values to account for multiple comparisons. Significance levels: *p<0.05. **p<0.01. ***p<0.001.

| Contrast | Odds ratio | SE | 95% CI | p-value |
|---|---|---|---|---|
| Averaging | 1.05 | 0.07 | [0.88, 1.25] | 0.93 |
| Experienced Bird Weighting | 1.13 | 0.08 | [0.94, 1.34] | 0.34 |
| Maximize Generation | 1.15 | 0.08 | [0.96, 1.37] | 0.19 |

*Appendix 1—table 5 Continued on next page*

*Appendix 1—table 5 Continued*

| Contrast | Odds ratio | SE | 95% CI | p-value | |
|---|---|---|---|---|---|
| Better Bird Weighting | 1.37 | 0.09 | [1.15, 1.63] | <0.001*** | *** |
| All-or-Nothing | 0.99 | 0.07 | [0.84, 1.19] | 1.00 | |
| Maximize Flight | 1.55 | 0.11 | [1.30, 1.86] | <0.001*** | *** |
| Maximize Time Steps | 1.26 | 0.08 | [1.06, 1.51] | <0.01** | ** |

**Appendix 1—table 6.** Contrasts between estimated social weights of the experienced bird for each strategy and the experimental data holding generation constant (at its mean value).
Contrasts were conducted on the response scale relative to the experimental data. CIs were calculated at the 95% interval. Dunnett's correction was applied to both CIs and p-values.
Significance levels: *p<0.05. **p<0.01. ***p<0.001.

| Contrast | Mean difference | SE | 95% CI | p-value |
|---|---|---|---|---|
| Averaging | 0.003 | 0.03 | [–0.08, 0.09] | 0.99 |
| Experienced Bird Weighting | 0.14 | 0.03 | [0.05, 0.23] | <0.001*** |
| Maximize Generation | 0.18 | 0.04 | [0.09, 0.27] | <0.001*** |
| Better Bird Weighting | 0.17 | 0.04 | [0.06, 0.29] | <0.001*** |
| All-or-Nothing | 0.23 | 0.04 | [0.11, 0.34] | <0.001*** |
| Maximize Flight | 0.23 | 0.04 | [0.11, 0.34] | <0.001*** |
| Maximize Time Steps | 0.12 | 0.04 | [0.00, 0.23] | 0.04* |

**Appendix 1—table 7.** Contrasts derived from the estimated marginal means (EMMs) of the beta regression model applied to the route efficiency data.
Contrasts were conducted on the log odds ratio scale relative to the experimental data. CIs were calculated at the 95% interval. Dunnett's correction was applied to both CIs and p-values.
Significance levels: *p<0.05. **p<0.01. ***p<0.001.

| Contrast | Generation | Odds ratio | SE | 95% CI | p-value |
|---|---|---|---|---|---|
| | 1 | 1.12 | 0.17 | [0.75, 1.67] | 0.91 |
| | 2 | 0.98 | 0.14 | [0.67, 1.43] | 1.00 |
| | 3 | 1.29 | 0.18 | [0.90, 1.85] | 0.30 |
| | 4 | 0.90 | 0.14 | [0.60, 1.35] | 0.94 |
| Averaging | 5 | 1.05 | 0.16 | [0.70, 1.58] | 0.99 |
| | 1 | 1.12 | 0.17 | [0.75, 1.67] | 0.91 |
| | 2 | 0.98 | 0.14 | [0.67, 1.43] | 1.00 |
| | 3 | 1.37 | 0.19 | [0.95, 1.96] | 0.13 |
| | 4 | 1.02 | 0.16 | [0.68, 1.53] | 1.00 |
| Experienced Bird Weighting | 5 | 1.23 | 0.19 | [0.82, 1.86] | 0.60 |
| | 1 | 1.12 | 0.17 | [0.75, 1.67] | 0.91 |
| | 2 | 0.98 | 0.14 | [0.67, 1.43] | 1.00 |
| | 3 | 1.40 | 0.19 | [0.97, 2.03] | 0.08 |
| | 4 | 1.08 | 0.17 | [0.71, 1.63] | 0.74 |
| Maximize Generation | 5 | 1.36 | 0.21 | [0.89, 2.07] | 0.22 |

*Appendix 1—table 7 Continued on next page*

*Appendix 1—table 7 Continued*

| Contrast | Generation | Odds ratio | SE | 95% CI | p-value |
|---|---|---|---|---|---|
| | 1 | 1.12 | 0.17 | [0.75, 1.67] | 0.91 |
| | 2 | 1.15 | 0.16 | [0.78, 1.69] | 0.83 |
| | 3 | **1.66** | **0.23** | **[1.16, 2.39]** | **<0.01 \*\*** |
| | 4 | 1.27 | 0.20 | [0.84, 1.90] | 0.48 |
| Better Bird Weighting | 5 | **1.57** | **0.24** | **[1.05, 2.37]** | **0.02 \*** |
| | 1 | 1.12 | 0.17 | [0.75, 1.67] | 0.91 |
| | 2 | 0.94 | 0.13 | [0.64, 1.38] | 1.00 |
| | 3 | 1.23 | 0.17 | [0.85, 1.78] | 0.14 |
| | 4 | 0.88 | 0.14 | [0.58, 1.34] | 0.89 |
| All-or-Nothing | 5 | 1.06 | 0.16 | [0.70, 1.61] | 0.99 |
| | 1 | 1.12 | 0.17 | [0.75, 1.67] | 0.91 |
| | 2 | 1.23 | 0.18 | [0.83, 1.79] | 0.54 |
| | 3 | **1.89** | **0.26** | **[1.32, 2.72]** | **<0.001 \*\*\*** |
| | 4 | **1.51** | **0.23** | **[1.00, 2.27]** | **0.046 \*** |
| Maximize Flight | 5 | **1.96** | **0.30** | **[1.30, 2.94]** | **<0.001 \*\*\*** |
| | 1 | 1.12 | 0.17 | [0.75, 1.67] | 0.91 |
| | 2 | 1.07 | 0.15 | [0.73, 1.57] | 0.76 |
| | 3 | **1.54** | **0.21** | **[1.07, 2.22]** | **0.01 \*** |
| | 4 | 1.18 | 0.18 | [0.79, 1.78] | 0.76 |
| Maximize Time Steps | 5 | 1.49 | 0.23 | [0.99, 2.24] | 0.06 |

**Appendix 1—table 8.** Contrasts derived from the estimated marginal means (EMMs) of the linear mixed model applied to the social weights of the experienced bird.

Contrasts were conducted on the response scale with mean difference relative to the experimental data. CIs were calculated at the 95% interval. Dunnett's correction was applied to both CIs and p-values. Significance levels: \*p<0.05. \*\*p<0.01. \*\*\*p<0.001.

| Contrast | Generation | Mean difference | SE | 95% CI | p-value |
|---|---|---|---|---|---|
| | 2 | –0.03 | 0.06 | [–0.19, 0.14] | 0.98 |
| | 3 | 0.05 | 0.06 | [–0.12, 0.22] | 0.92 |
| | 4 | –0.05 | 0.07 | [–0.24, 0.14] | 0.95 |
| Averaging | 5 | 0.04 | 0.08 | [–0.17, 0.24] | 0.98 |
| | 2 | 0.11 | 0.06 | [–0.05, 0.28] | 0.38 |
| | 3 | **0.19** | **0.06** | **[0.02, 0.38]** | **0.02 \*** |
| | 4 | 0.09 | 0.07 | [–0.09, 0.28] | 0.64 |
| Experienced Bird Weighting | 5 | 0.17 | 0.08 | [–0.03, 0.38] | 0.13 |
| | 2 | –0.03 | 0.06 | [–0.20, 0.14] | 0.98 |
| | 3 | **0.22** | **0.06** | **[0.04, 0.39]** | **<0.01 \*\*** |
| | 4 | **0.20** | **0.08** | **[0.01, 0.39]** | **0.03 \*** |
| Maximize Generation | 5 | **0.33** | **0.08** | **[0.13, 0.54]** | **<0.001 \*\*\*** |

*Appendix 1—table 8 Continued on next page*

*Appendix 1—table 8 Continued*

| Contrast | Generation | Mean difference | SE | 95% CI | p-value |
|---|---|---|---|---|---|
| | 2 | –0.03 | 0.08 | [–0.23, 0.17] | 0.99 |
| | 3 | **0.22** | **0.08** | **[0.00, 0.44]** | **0.05 *** |
| Better Bird Weighting | 4 | 0.19 | 0.09 | [–0.04, 0.43] | 0.17 |
| | 5 | **0.29** | **0.09** | **[0.05, 0.54]** | **0.01 *** |
| | 2 | –0.03 | 0.08 | [–0.26, 0.20] | 0.99 |
| | 3 | **0.26** | **0.08** | **[0.03, 0.48]** | **0.01 *** |
| | 4 | **0.26** | **0.09** | **[0.02, 0.49]** | **0.02 *** |
| All-or-Nothing | 5 | **0.39** | **0.09** | **[0.14, 0.64]** | **<0.001 *** |
| | 2 | –0.03 | 0.09 | [–0.26, 0.20] | 0.99 |
| | 3 | **0.27** | **0.08** | **[0.04, 0.49]** | **0.01 *** |
| | 4 | **0.26** | **0.09** | **[0.05, 0.49]** | **0.03 *** |
| Maximize Flight | 5 | **0.39** | **0.09** | **[0.14, 0.63]** | **<0.001 *** |
| | 2 | –0.03 | 0.09 | [–0.26, 0.20] | 0.99 |
| | 3 | 0.17 | 0.08 | [–0.05, 0.39] | 0.23 |
| Maximize Time Steps | 4 | 0.14 | 0.09 | [–0.10, 0.37] | 0.48 |
| | 5 | **0.26** | **0.09** | **[0.00, 0.50]** | **0.04 *** |

**Appendix 1—table 9.** Bootstrap contrasts for different strategies relative to the experimental control for the route efficiency data.
The table reports odds ratios, 95% CIs and adjusted p-values. Multiple comparison adjustments were made using Dunnett's method. Significance levels: *p<0.05. **p<0.01. ***p<0.001.

| Contrast | Generation | Odds ratio | 95% CI | p-value |
|---|---|---|---|---|
| | 1 | 1.09 | [0.51, 2.35] | 1.0 |
| | 2 | 1.01 | [0.49, 2.07] | 1.0 |
| | 3 | 1.47 | [0.72, 3.00] | 0.53 |
| | 4 | 0.86 | [0.45, 1.65] | 0.95 |
| Averaging | 5 | 1.01 | [0.48, 2.17] | 1.0 |
| | 1 | 1.08 | [0.51, 2.33] | 1.0 |
| | 2 | 0.97 | [0.46, 2.04] | 1.0 |
| | 3 | 1.54 | [0.76, 3.09] | 0.42 |
| | 4 | 0.97 | [0.54, 1.76] | 1.0 |
| Experienced Bird Weighting | 5 | 1.21 | [0.61, 2.39] | 0.92 |
| | 1 | 1.09 | [0.51, 2.34] | 1.0 |
| | 2 | 1.01 | [0.49, 2.06] | 1.0 |
| | 3 | 1.58 | [0.79, 3.14] | 0.35 |
| | 4 | 1.02 | [0.58, 1.79] | 1.0 |
| Maximize Generation | 5 | 1.32 | [0.70, 2.50] | 0.71 |

*Appendix 1—table 9 Continued on next page*

*Appendix 1—table 9 Continued*

| Contrast | Generation | Odds ratio | 95% CI | p-value |
|---|---|---|---|---|
| | 1 | 1.09 | [0.51, 2.34] | 1.0 |
| | 2 | 1.20 | [0.60, 2.43] | 0.93 |
| | 3 | **1.96** | **[1.08, 3.55]** | **0.01 *** |
| Better Bird Weighting | 4 | 1.26 | [0.83, 1.91] | 0.52 |
| | 5 | 1.61 | [0.96, 2.69] | 0.08 |
| | 1 | 1.09 | [0.51, 2.35] | 1.0 |
| | 2 | 0.99 | [0.52, 1.89] | 1.0 |
| | 3 | 1.44 | [0.82, 2.51] | 0.38 |
| | 4 | 0.87 | [0.61, 1.22] | 0.74 |
| All-or-Nothing | 5 | 1.07 | [0.69, 1.65] | 0.99 |
| | 1 | 1.09 | [0.51, 2.34] | 1.0 |
| | 2 | 1.12 | [0.62, 2.65] | 0.85 |
| | 3 | **2.21** | **[1.19, 4.12]** | **<0.01 *** |
| | 4 | **1.51** | **[1.00, 2.27]** | **0.047 *** |
| Maximize Flight | 5 | **2.01** | **[1.26, 3.23]** | **<0.001 *** |
| | 1 | 1.09 | [0.51, 2.34] | 1.0 |
| | 2 | 1.16 | [0.55, 2.27] | 0.99 |
| | 3 | 1.80 | [0.96, 3.36] | 0.08 |
| | 4 | 1.16 | [0.74, 1.80] | 0.87 |
| Maximize Time Steps | 5 | 1.49 | [0.89, 2.50] | 0.20 |

**Appendix 1—table 10.** Bootstrap mean difference contrasts for different strategies relative to the experimental control for the social weights data.
The table reports mean differences, 95% CIs, and adjusted p-values using Dunnett's method.
Significance levels: **p<0.05. **p<0.01. ***p<0.001.

| Contrast | Generation | Mean difference | 95% CI | p-value |
|---|---|---|---|---|
| | 2 | –0.03 | [–0.24, 0.18] | 0.99 |
| | 3 | 0.05 | [–0.14, 0.23] | 0.93 |
| | 4 | –0.05 | [–0.34, 0.24] | 0.98 |
| Averaging | 5 | 0.04 | [–0.24, 0.31] | 0.99 |
| | 2 | 0.11 | [–0.10, 0.32] | 0.58 |
| | 3 | **0.19** | **[0.01, 0.37]** | **0.041 *** |
| | 4 | 0.09 | [–0.20, 0.38] | 0.88 |
| Experienced Bird Weighting | 5 | 0.18 | [–0.10, 0.45] | 0.37 |
| | 2 | –0.03 | [–0.24, 0.18] | 0.99 |
| | 3 | **0.22** | **[0.03, 0.49]** | **0.01 *** |
| | 4 | 0.20 | [–0.08, 0.49] | 0.29 |
| Maximize Generation | 5 | **0.34** | **[0.06, 0.61]** | **<0.01 *** |

*Appendix 1—table 10 Continued on next page*

*Appendix 1—table 10 Continued*

| Contrast | Generation | Mean difference | 95% CI | p-value |
|---|---|---|---|---|
| | 2 | –0.03 | [–0.36, 0.31] | 1.00 |
| | 3 | 0.22 | [0.00, 0.49] | 0.16 |
| | 4 | 0.19 | [–0.13, 0.52] | 0.47 |
| Better Bird Weighting | **5** | **0.30** | **[0.06, 0.61]** | <0.01 ** |
| | 2 | –0.03 | [–0.52, 0.47] | 1.00 |
| | 3 | 0.26 | [–0.17, 0.68] | 0.44 |
| | 4 | 0.26 | [–0.21, 0.73] | 0.53 |
| All-or-Nothing | 5 | 0.40 | [–0.04, 0.84] | 0.09 |
| | 2 | –0.03 | [–0.32, 0.26] | 1.00 |
| | **3** | **0.27** | **[0.01, 0.53]** | 0.04 * |
| | 4 | 0.26 | [–0.07, 0.59] | 0.21 |
| Maximize Flight | **5** | **0.39** | **[0.08, 0.70]** | <0.01 ** |
| | 2 | –0.03 | [–0.35, 0.29] | 1.00 |
| | 3 | 0.16 | [–0.12, 0.45] | 0.46 |
| Maximize Time Steps | 4 | 0.14 | [–0.23, 0.50] | 0.82 |
| | 5 | 0.26 | [–0.10, 0.61] | 0.28 |

