## [Editor Report · eLife Assessment]

This study addresses an **important** question and shows how social navigation in homing pigeons can be explained by simple averaging, without requiring any complex cognitive abilities. The evidence, based on a rigorous and systematic comparison of seven models and data on how social routes can be generated from solitary routes, is **compelling**. The authors should be commended for their willingness to critically re-examine established interpretations.

---

## [Referee Report · Reviewer #1 (Public review)]

Summary:

This study investigates how collective navigation improvements arise in homing pigeons. Building on the Sasaki & Biro (2017) experiment on homing pigeons, the authors use simulations to test seven candidate social learning strategies of varying cognitive complexity, ranging from simple route averaging to potentially cognitively demanding selective propagation of superior routes. They show that only the simplest strategy-equal route averaging-quantitatively matches the experimental data in both route efficiency and social weighting. More complex strategies, while potentially more effective, fail to align with the observed data. The authors also introduce the concept of "effective group size," showing that the chaining design leads to a strong dilution of earlier individuals' contributions. Overall, they conclude that cognitive simplicity rather than cumulative cultural evolution explains collective route improvements in pigeons.

Strengths:

The manuscript provides a compelling argument that a simpler hypothesis is necessary and sufficient to explain the findings of a recent study on improvements to pigeon routes, through a rigorous, systematic comparison of seven alternative hypotheses. The authors should be commended for their willingness to critically re-examine established interpretations. The introduction and discussion are broad and link pigeon navigation to general debates on social learning, wisdom of crowds, and CCE.

Weaknesses:

The authors' method focuses on trajectory-level average behaviour rather than the fine-scale decision-making processes of organisms. This is acknowledged in the manuscript by the authors.

Comments on revision:

The authors have addressed most of the comments by me as well as the other reviewer.

---

## [Referee Report · Reviewer #2 (Public review)]

Summary:

The manuscript investigates which social navigation mechanisms, with different cognitive demands, can explain experimental data collected from homing pigeons. Interestingly, the results indicate that the simplest strategy - route averaging - aligns best with the experimental data, while the most demanding strategy - selectively propagating the best route - offers no advantage. Further, the results suggest that a mixed strategy of weighted averaging may provide significant improvements.

The manuscript addresses the important problem of identifying possible mechanisms that could explain observed animal behavior by systematically comparing different candidate models. A core aspect of the study is the calculation of collective routes from individual bird routes using different models that were hypothesized to be employed by the animals but which differ in their cognitive demands.

The manuscript is well written, with high-quality figures supporting both the description of the approach taken and the presentation of results. The results should be of interest to a broad community of researchers investigating (collective) animal behavior, ranging from experiment to theory. The general approach and mathematical methods appear reasonable and show no obvious flaws. The statistical methods also appear.

Strengths:

The main strength of the manuscript is the systematic comparison of different meta-mechanisms for social navigation by modeling social trajectories from solitary trajectories and directly comparing them with experimental results on social navigation. The results show that the experimentally observed behavior could, in principle, arise from simple route averaging without the need to identify "knowledgeable" individuals. Another strength of the work is the establishment of a connection between social navigation behavior and the broader literature on the wisdom of crowds through the concept of effective group size.

Comments on revision:

The authors made substantial revisions to the manuscript, addressing my comments. While I do think that regarding my second comment on CCE the authors could be a bit more bold, I am overall satisfied with the revisions made.

---

## [Author Response]

The following is the authors’ response to the original reviews.

**Public Reviews:**

**Reviewer #1 (Public review):**
Summary:This study investigates how collective navigation improvements arise in homing pigeons. Building on the Sasaki & Biro (2017) experiment on homing pigeons, the authors use simulations to test seven candidate social learning strategies of varying cognitive complexity, ranging from simple route averaging to potentially cognitively demanding selective propagation of superior routes. They show that only the simplest strategy-equal route averaging-quantitatively matches the experimental data in both route efficiency and social weighting. More complex strategies, while potentially more effective, fail to align with the observed data. The authors also introduce the concept of "effective group size," showing that the chaining design leads to a strong dilution of earlier individuals' contributions. Overall, they conclude that cognitive simplicity rather than cumulative cultural evolution explains collective route improvements in pigeons.Strengths:The manuscript addresses an important question and provides a compelling argument that a simpler hypothesis is necessary and sufficient to explain findings of a recent influential study on pigeon route improvements, via a rigorous systematic comparison of seven alternative hypotheses. The authors should be commended for their willingness to critically re-examine established interpretations. The introduction and discussion are broad and link pigeon navigation to general debates on social learning, wisdom of crowds, and CCE.

We thank the reviewer for their positive comments.

Weaknesses:The lack of availability of codes and data for this manuscript, especially given that it critically examines and proposes alternative hypotheses for an important published work.

We thank the reviewer for their comment. The code and data for our manuscript are an important aspect of the study, and we had intended to make them publicly available upon publication. The link to our code and data on fig share can be found here: (https://doi.org/10.6084/m9.figshare.28950032.v1). We have now revised the manuscript to include a link to our dataset.

**Reviewer #2 (Public review):**
Summary:The manuscript investigates which social navigation mechanisms, with different cognitive demands, can explain experimental data collected from homing pigeons. Interestingly, the results indicate that the simplest strategy - route averaging - aligns best with the experimental data, while the most demanding strategy - selectively propagating the best route - offers no advantage. Further, the results suggest that a mixed strategy of weighted averaging may provide significant improvements.The manuscript addresses the important problem of identifying possible mechanisms that could explain observed animal behavior by systematically comparing different candidate models. A core aspect of the study is the calculation of collective routes from individual bird routes using different models that were hypothesized to be employed by the animals, but which differ in their cognitive demands.The manuscript is well-written, with high-quality figures supporting both the description of the approach taken and the presentation of results. The results should be of interest to a broad community of researchers investigating (collective) animal behavior, ranging from experiment to theory. The general approach and mathematical methods appear reasonable and show no obvious flaws. The statistical methods also appear.Strengths:The main strength of the manuscript is the systematic comparison of different meta-mechanisms for social navigation by modeling social trajectories from solitary trajectories and directly comparing them with experimental results on social navigation. The results show that the experimentally observed behavior could, in principle, arise from simple route averaging without the need to identify "knowledgeable" individuals. Another strength of the work is the establishment of a connection between social navigation behavior and the broader literature on the wisdom of crowds through the concept of effective group size.

We thank the reviewer for their positive comments.

Weaknesses:However, there are two main weaknesses that should be addressed:(1) The first concerns the definition of "mechanism" as used by the authors, for example, when writing "navigation mechanism." Intuitively, one might assume that what is meant is a behavioral mechanism in the sense of how behavior is generated as a dynamic process. However, here it is used at a more abstract (meta) level, referring to high-level categories such as "averaging" versus "leader-follower" dynamics. It is not used in the sense of how an individual makes decisions while moving, where the actual route followed in a social context emerges from individuals navigating while simultaneously interacting with conspecifics in space and time. In the presented work, the approach is to directly combine (global) route data of solitary birds according to the considered "meta-mechanisms" to generate social trajectories. Of course, this is not how pigeon social navigation actually works-they do not sit together before the flight and say, "This is my route, this is your route, let's combine them in this way." A mechanistic modeling approach would instead be some form of agent-based model that describes how agents move and interact in space and time. Such a "bottom-up" approach, however, has its drawbacks, including many unknown parameters and often strongly simplifying (implicit) assumptions. I do not expect the authors to conduct agent-based modeling, but at the very least, they should clearly discuss what they mean by "mechanism" and clarify that while their approach has advantages-such as naturally accounting for the statistical features of solitary routes and allowing a direct comparison of different meta-mechanisms is also limited, as it does not address how behavior is actually generated. For example, the approach lacks any explicit modeling of errors, uncertainty, or stochasticity more broadly (e.g., due to environmental influences). Thus, while the presented study yields some interesting results, it can only be considered an intermediate step toward understanding actual behavioral mechanisms.

We thank the reviewer for their comment and thoughtful suggestions. We agree that the inherent behavioral mechanisms and the biological basis of these mechanisms cannot be determined just through the navigational data alone. For instance, it remains unexplored if pigeons are adapting their behavior based only on social cues from their partners or using other navigational features such as landmarks or roads, location of the sun, geomagnetic cues or prior learnt routes. However, we do agree (as also pointed by the reviewer) that these behavioral rules generate an emergent ‘meta-mechanism’ where the bird pairs are behaving as if their preferred routes are averaged during a flight. It will be important in future work to explore the biological basis of these mechanisms, but our current approach allows us to only describe the mechanisms in a meta sense with any confidence. Considering this, we believe that our analysis is a more top-down approach towards describing the outcomes of these underlying mechanisms in an abstract sense. We would also like to point the reviewer to Dalmaijer, 2024 [1] who used a bottom up approach, using naive agents and showed that cumulative route improvements emerged in the absence of any sophisticated communication in the same dataset, in agreement with our approach. We have now added a paragraph: “It is also important to clarify that we use the terms…… that lead to these meta-mechanisms arising remain an open question.” found in lines 120-129 in our Introduction to make this clarification.

(2) While the presented study raises important questions about the applicability and viability of cumulative cultural evolution (CCE) in explaining certain animal behaviors such as social navigation, I find that it falls short in discussing them. What are the implications regarding the applicability of CCE to animal data and to previously claimed experimental evidence for CCE? Should these experiments be re-analyzed or critically reassessed? If not, why? What are good examples from animal behavior where CCE should not be doubted? Furthermore, what about the cited definitions and criteria of CCE? Are they potentially too restrictive? Should they be revised-and if so, how? Conversely, if the definitions become too general, is CCE still a useful concept for studying certain classes of animal behavior? I think these are some of the very important questions that could be addressed or at least raised in the discussion to initiate a broader debate within the community.

We thank the reviewer for their comments and interesting questions regarding our study. We agree with the reviewer that our study opens up new avenues for critically analysing the criteria previous studies have used for providing evidence of CCE in non-human animals. According to our literature review, we found that the field has been usually motivated in thinking about CCE in a ‘process’ focused manner (Reindl et al. [2]) in regards to individuals being able to compare strategies and selecting ones resulting in higher individual fitness. This preferential selection of strategies – termed innovations — allows for the stereotypical ratcheting effect seen in CCE. In our study, we propose that in the case of homing pigeons, the ratcheting effect is more of a statistical outcome rather than deliberate individual judgement. We believe that this strategy is also amenable to certain task types (which in our study was homing route choice) and may change for others (for example solving a puzzle box) and the task also needs to be sufficiently complex for animals to benefit from the use of social information (Caldwell et al. 2008 [3]). Thus, we recommend future work to address what classes of problems would fit well within the definition of “emergent” CCE and which ones don’t. Keeping this framework in mind, studies should clearly state what definition of CCE they are using and should be critically evaluated for their underlying task type and cognitive mechanisms to deem them as CCE. Considering these points, we have now expanded our Discussion to include a paragraph: “Our results highlight the need for more…..range of task types and cognitive abilities.” found in lines 420-433 to highlight these key questions.

**Recommendations for the authors:**

**Reviewer #1 (Recommendations for the authors):**
I do not have any major objections, but I am clarifying my points as major or minor depending on the effort required to address (mostly via rewriting and clarifications).Major comments:(1) A schematic summary of the original study: Since the current manuscript builds directly on Sasaki & Biro (2017), it would greatly help readers if you included a concise schematic figure summarizing the original experiment. For instance, a simple panel could depict the chain design (experienced + naïve replacements), the control treatments, and the key empirical findings (improvements in route efficiency across generations, and route similarity within vs. between chains). Presenting this visually would save readers the effort of reconstructing the design and main results from text alone, especially for those unfamiliar with the original paper. It would also clarify exactly what empirical patterns your simulations are intended to reproduce.

We thank the reviewer for this comment. We have now revised the manuscript with a schematic illustration adapted from the original study by Sasaki and Biro (2017). We hope this clarifies the experimental design and results we aimed to highlight in our work.

(2) Reproducibility: Code and data are only "available on request." I believe eLife has strong policies on open science; a lack of immediate open access to analysis would be a barrier. I find it jarring that a paper intending to reproduce and improvise a previously published paper does not make the codes and data available for peer review or to readers without an explicit request.

We have taken the feedback into consideration and updated the Data Availability section with a link to our Fig share dataset.

(3) One huge drawback of the current format of the manuscript, where Methods come after Results, is that one has to really struggle to understand and appreciate Figures 2 and 3. I would strongly urge authors to have a shorter methods section embedded either as a subsection before the Results, or within the results section, as described in each figure. Perhaps a lot of my confusion also comes from not having known the previous paper, but it may be true for other readers, too. More specifically, for Figure 3, how is social weight for the experiments inferred? Figure 3 caption talks of mean difference, but one has to check the manuscript at multiple places throughout to really understand what this difference is (the definition) and how it is computed.

While we agree that our manuscript includes the Methods section at the end, we tried to structure our text to tell a story (as stated in our manuscript title). To this end, we organized the text into short titled subsections that briefly convey the relevant background, identify the knowledge gap and outline our approach. We chose this structure to reserve the indepth details about model implementation and statistical analysis for the Methods.

Additionally, we made sure to include references to methodological details in relevant segments of the Introduction and Results section so as to not bog down the reader by model complexities and keep a coherent narrative that delivers the message of our study. To further address the background of our work, we have now added a schematic of the original study in response to a previous comment by the reviewer, which we hope helps the reader better understand our work. We hope this explanation clarifies the intention behind our writing choice and decision to retain the current structure.

(4) The introduction of the 'effective group size' concept is a potentially valuable and intuitive way to interpret chain dynamics, but the explanation is somewhat buried in the Results/Methods; I suggest highlighting it more prominently (e.g., in the Discussion or with a schematic in the Results) so readers can readily grasp this useful idea.

We thank the reviewer that they found our concept of ‘effective group size’ useful. However, we do believe that we introduced the idea and rationale behind using this method in the Results: “We asked to what extent……to an equivalent group size” found in lines 305-314. We reserved a detailed description of this method in the Methods section. However, to further emphasize the importance of the concept we have now added a text: “This is further supported….. slightly better than two individuals.” found in lines 389-394 in the Discussion.

Minor comments:(1) Line 12: "what is the navigation mechanism(s)" - the (s) is a bit awkward. Either remove (s) or ask what the mechanisms are.

We have fixed the typo to clarify the statement.

(2) Line 78: "Such 'ratchet'-like improvements is referred to..." → "are referred to."

We have fixed the typo to clarify the statement.

(3) Figure 3 caption: "color scheme in the plots are same" → should be "is the same."

We have fixed the typo to clarify the statement.

(4) Clarification on reporting confidence intervals: The manuscript reports confidence intervals (CIs) for the model-based comparisons (e.g., Figures 2-3). This might seem unnecessary for simulation studies, since running more iterations can arbitrarily shrink uncertainty. However, in your case, the CIs are justified because the simulations are anchored to a finite empirical dataset (only 9 solo trajectories), sampled with replacement, and analyzed with mixed-effects models that incorporate bird identity as a random effect. Thus, the intervals reflect biological sample variability rather than simulation noise. This must be clarified.

We have added a clarifying statement: “...and reflect the biological uncertainty in the empirical dataset, not simulation noise” found in lines 241 and 293 in the captions of Figures 2 and 3 in accordance with the reviewer’s comment.

(5) One part of the issue is that details of methods come much later in the manuscript, perhaps following journal style. Therefore, I recommend explicitly highlighting this rationale in the Results, so readers do not misinterpret the CIs as simply reflecting simulation error.

We believe that the clarifying statements we have now added in the captions of Figures 2 and 3 should convey this interpretation of CIs and further changes in the Results may not be required.

With these proposed changes we hope that we improved upon the clarity of our manuscript.

References:

(1) Dalmaijer ES (2024) Cumulative route improvements spontaneously emerge in artificial navigators even in the absence of sophisticated communication or thought. PLoS Biol. 22:e3002644.

(2) Reindl, E., Gwilliams, A.L., Dean, L.G. et al. (2020) Skills and motivations underlying children’s cumulative cultural learning: case not closed. Palgrave Commun 6, 106.

(3) Caldwell CA, Millen AE (2008) Studying cumulative cultural evolution in the laboratory. Phil. Trans. R. Soc. B 363:3529-3539.